# Biocontrol Potential of *Bacillus stercoris* Strain DXQ-1 Against Rice Blast Fungus Guy11

**DOI:** 10.3390/microorganisms13071538

**Published:** 2025-06-30

**Authors:** Qian Xu, Zhengli Shan, Zhihao Yang, Haoyu Ma, Lijuan Zou, Ming Dong, Tuo Qi

**Affiliations:** Ecological Security and Protection Key Laboratory of Sichuan Province, Mianyang Teachers’ College, Mianyang 621000, China; xuqian200020@163.com (Q.X.); x2110350518@163.com (Z.S.); yangzh0220@126.com (Z.Y.); myy17596550653@163.com (H.M.); ljzou66@163.com (L.Z.); dongming@hznu.edu.cn (M.D.)

**Keywords:** *Bacillus*, fungal pathogen, plant disease control, *Magnaporthe oryzae*, antifungal metabolites

## Abstract

Fungal diseases severely threaten global agriculture, while conventional chemical fungicides face increasing restrictions due to environmental and safety concerns. In this study, we isolated a soil-derived *Bacillus stercoris* strain, DXQ-1, exhibiting strong antagonistic activity against plant pathogenic fungi, notably *Magnaporthe oryzae*, the causal agent of rice blast. Scanning electron microscopy revealed that DXQ-1 disrupts fungal hyphae and inhibits conidial germination, with a 24 h crude broth treatment reducing germination to 83.33% and completely blocking appressoria formation. LC-MS-based metabolomic analysis identified key antifungal components, including lipids (35.83%), organic acid derivatives (22.15%), and small bioactive molecules (e.g., Leu-Pro, LPE 15:0). After optimizing fermentation conditions (LB medium, pH 7.0, 28 °C, 48 h), the broth showed >90% inhibition against *M. oryzae* and *Nigrospora oryzae* and retained high thermal (68 °C, 1 h) and UV (4 h) stability. Field trials demonstrated effective disease control and significant promotion of rice growth, increasing plant height (17.7%), fresh weight (53.3%), and dry weight (33.3%). These findings highlight DXQ-1 as a promising biocontrol agent, offering a sustainable and effective alternative for managing fungal diseases in crops.

## 1. Introduction

Fungal diseases represent a major threat to global crop production, while the extensive use of chemical fungicides is increasingly constrained by environmental pollution, resistance development, and food safety issues. This has driven the urgent demand for sustainable, eco-friendly alternatives. Among them, *Bacillus* species have emerged as promising biocontrol agents due to their broad-spectrum antifungal activity, plant-growth-promotion capabilities, and environmental resilience. *Bacillus* strains suppress pathogens through multiple mechanisms, including the secretion of lipopeptides (e.g., iturin, surfactin) [1,2,3], chitinases [4], and volatile organic compounds [5], as well as by inducing systemic resistance in host plants [6]. Recent studies have demonstrated the potential of various *Bacillus* strains, such as *B. velezensis* and *B. thuringiensis*, in inhibiting a wide range of pathogens through metabolite production and endophytic colonization [4,6,7] Moreover, synergistic effects have been observed when combining *Bacillus* with chemical fungicides, enhancing disease control and improving crop yield [6]. Despite progress in understanding *Bacillus* antifungal mechanisms, key challenges remain, including identifying functional genes related to antifungal activity, elucidating the complex interactions among microbes, plants, and pathogens, and ensuring efficacy under field conditions [8,9,10]. Therefore, further research into the functional genomics, strain screening, and field stability of *Bacillus*-based biocontrol agents is essential.

*Bacillus* species have gained increasing attention in plant disease management due to their broad-spectrum antifungal activity, environmental adaptability, and dual functionality in pathogen suppression and plant growth promotion. Recent studies highlight their potential as eco-friendly biocontrol agents. For instance, Endophytic *Bacillus* strains isolated from coffee leaves demonstrating both antifungal activity against soil-borne fungi and insecticidal effects on Lepidoptera pests [11]. Whole-genome sequencing of *B. velezensis* BN revealed 12 secondary metabolite biosynthetic gene clusters—including those for surfactin, iturin, and fengycin—whose synergistic action conferred broad resistance to pathogens like *Fusarium* spp. and *Botrytis cinereal* [3]. Additionally, genes encoding glycoside hydrolases and chitinases were found to enhance antifungal efficacy. Beyond disease suppression, *Bacillus* strains also promote plant growth. Rhizosphere-derived *B. velezensis* improved maize seedling growth by producing 3S,4R-ABD and 3R,4R-ABD, while nonvolatile compounds accumulating in the root maturation zone may regulate the relationship between plant roots and beneficial bacteria. Co-inoculation of *Bacillus* with other beneficial microorganisms can also result in niche complementarity, thereby enhancing the stability of biological controls [12,13,14,15]. Despite these advances, challenges remain in understanding the molecular mechanisms underlying metabolite biosynthesis and field efficacy.

*Bacillus* species exhibit broad-spectrum antimicrobial activity through the secretion of secondary metabolites that disrupt the cellular structures of plant pathogens [5,10]. In addition, they can trigger host defense responses by enhancing the activity of disease-resistance-related enzymes and by producing volatile organic compounds (VOCs) that inhibit pathogen growth and development [16,17,18,19]. Moreover, *Bacillus* functions as a plant-growth-promoting rhizobacteria (PGPR), playing a crucial role in crop protection, stimulating plant growth, and improving soil health [20,21,22,23]. Building on previous research, this study reports the isolation of a highly effective *B. stercoris* strain, designated DXQ-1. This study identifies and characterizes a soil-derived *B. stercoris* strain, DXQ-1, with potent antifungal activity and plant-growth-promoting effects, highlighting its potential as a sustainable biocontrol agent against rice fungal diseases. Our findings provide a scientific foundation for the development of robust, sustainable biocontrol strategies for integrated plant disease management.

## 2. Materials and Methods

### 2.1. Isolation and Identification of DXQ-1

In a rice field with a high incidence of rice blast disease, a patch of healthy rice plants was identified. Fresh soil was passed through a 20-mesh sieve, and 5 g was weighed and placed into a pre-prepared Erlenmeyer flask. Then, 45 mL of sterile water and an appropriate amount of glass beads were added. The mixture was shaken at 200 rpm/min at a constant temperature of 25 °C for 30 min to obtain a soil suspension. Using the gradient dilution method, three concentration gradients of the soil suspension (10^−1^, 10^−2^, 10^−3^) were prepared with sterile water in a laminar flow hood. For each gradient, 200 μL of the soil suspension was evenly spread on the surface of Gauze’s synthetic broth medium, Luria-Bertani culture plate and Nutrient Agar (NA) medium, and Potato Dextrose Agar (PDA) medium. Each treatment was performed in ten plates. To further screen for *Bacillus* species, the soil suspension was heated at 60 °C to induce spore formation and then spread onto NA medium. The plates were incubated upside–down at 28 °C for 3 days. Colonies with different morphologies were picked, streaked for purification, and numbered. The purified single colonies were transferred into 50% glycerol and stored at −80 °C for future use. We isolated over 300 colonies, including fungi, actinomycetes, *Bacillus* spp., etc. Using *M. oryzae* Guy11 as the indicator strain, biocontrol bacteria with antagonistic activity were preliminarily screened. After rescreening, the biocontrol *Bacillus* spp. with antagonistic effects were assigned corresponding identification numbers. The purified single colonies were preserved in a storage solution containing 50% glycerol and stored at −80 °C for future use.

To screen for bacterial strains with strong antagonistic activity against the rice blast pathogen, a dual culture assay was first performed. A mycelial plug was placed at the center of a PDA plate, and candidate *Bacillus* strains were streaked on opposite sides, 3 cm from the plug. Plates were incubated at 28 °C for 7 days, and inhibition zones were measured. Strains showing significant inhibition were further evaluated using the drug-amended plate method. Fermentation broths were centrifuged, and the supernatant was filtered (0.22 μm) to obtain sterile extracts. These were mixed with PDA at 10% (*v*/*v*) to prepare drug-containing media. A 5 mm mycelial plug was then inoculated at the center, and pathogen growth was assessed after incubation. The PDB mixed with PDA at 10% (*v*/*v*) was inoculated with *M. oryzae* strain Guy11 at the center, simultaneously with the control treatment. All treatments were performed in triplicate.

The biocontrol bacterium was discovered unintentionally during the cultivation of *M. oryzae* on CM medium, where a clear inhibition zone was observed. To identify the strain, 16S rRNA was amplified using specific primers (27F: AGA GTTTGATCMTGGCTCAG; 1492R: CGGTTACCTTGTTACGACTT) [24]. The resulting sequence was submitted to GenBank under accession number PV789457. Phylogenetic analysis was performed using the neighbor-joining (NJ) method in MEGA 7.0 software (RRID:SCR_011920) (https://www.megasoftware.net, accessed on 21 May 2024) [25].

### 2.2. Antifungal Activity and Pathogen Infection Experiments

Plant pathogenic fungi, including *Magnaporthe oryzae* (Guy11), *Rhizoctonia solani* (AG-1), *Bipolaris maydis* (C4), *Fusarium graminearum* (PH-1), and *Nigrospora oryzae* (LW11), were collected and are preserved in our laboratory. In this study, *M. oryzae* strain Guy11 was used as the indicator strain. Guy11 was cultured on CM medium at 28 °C for 5 days, and its spores were subsequently induced in tomato–oat medium at 28 °C for 8 days.

*M. oryzae* strain Guy11 was cultured on CM medium under alternating light and dark conditions at 28 °C for 7 days. After sufficient sporulation, 2 mL of sterile distilled water was added to the plate, and spores were gently dislodged from the colony surface using a pipette tip. The suspension was filtered through Miracloth to remove mycelial fragments and debris and then centrifuged at 8000 rpm for 3 min. The supernatant was discarded, and the washing step was repeated twice to obtain purified spores.

To observe conidial germination and appressorium formation, 30 μL of spore suspension was placed on a clean, hydrophobic slide and incubated in a humid chamber at 28 °C. Then, 5–7 cm rice leaves (cv. LTH) at the two-leaf stage were wounded, treated with 6-BA (1 mg/L, pH 7.0), and inoculated with *M. oryzae* spore suspension (2 × 10^5^ conidia/mL) mixed with fermentation broth or sterile water. Inoculated leaves were incubated in darkness for 12 h and then under a 12 h light/dark cycle for 5–7 days. Lesion number and length were recorded and analyzed using ImageJ software version 1.64r [26]. Microscopic observations were performed using an Olympus optical microscope. For leaf inoculation in field conditions, rice plants (cv. TP309) at the two-leaf stage were used. The fermentation broth was mixed with the spore suspension, and Tween-20 was added to a final concentration of 0.01%. The mixture was then sprayed onto rice seedlings that had been growing in the field for about 14 days. Disease incidence was recorded one week after inoculation. Each treatment was performed in triplicate.

### 2.3. Optimization of Fermentation Conditions for Strain DXQ-1

To optimize the fermentation conditions of strain DXQ-1, NB, LB, and PDB media were initially tested, followed by the replacement of various carbon and nitrogen sources and nutrients. The optimal medium was screened using the drug-amended plate method. Seed cultures (2% inoculum) were incubated in sterilized liquid media at 28 °C and 180 rpm for 48 h to obtain fermentation broth. The resulting filtrate was mixed with PDA at a 10% (*v*/*v*) ratio to prepare drug-amended plates. PDA without filtrate served as the control. Mycelial plugs (5 mm) of *M. oryzae* and *N. oryzae* were inoculated at the center of the plates. Six plates were incubated at 28 °C for 7 days, and colony diameters were measured. Each treatment was performed in triplicate.

To determine the growth curve, DXQ-1 was cultured in LB (2% inoculum) under shaking conditions (28 °C, 180 rpm). Optical density (OD_600_) was recorded every 12 h for 144 h. For fermentation time optimization, samples were collected at 24, 48, 72, 96, and 120 h. Antimicrobial activity of the fermentation filtrates was evaluated using the drug-amended plate method.

To assess the effect of initial pH, LB media adjusted to pH 3~10 were inoculated and fermented under standard conditions. Similarly, to evaluate temperature effects, fermentation was conducted at 18 °C, 23 °C, 28 °C, 33 °C, and 38 °C. All filtrates were tested for bioactivity using the drug-amended plate method, with three replicates per treatment.

### 2.4. Isolation of Antifungal Active Substances

Strain DXQ-1 was cultured in sterilized LB liquid medium (2% inoculum) at 28 °C and 180 rpm for 48 h to obtain the fermentation broth. The broth was centrifuged at 8000 rpm for 5 min, and the supernatant was collected for further extraction.

The supernatant underwent sequential solvent extraction using petroleum ether, dichloromethane, ethyl acetate, and n-butanol in a 1:1 (*v*/*v*) ratio, following a polarity gradient. After phase separation, both aqueous and organic layers were concentrated to dryness via rotary evaporation. Each dried extract was re-dissolved in 100 mL of distilled water and sterilized through a 0.22 μm membrane filter. The antifungal activity of each extract was evaluated using the drug-amended plate method.

To further investigate the presence and diffusibility of antifungal metabolites, a sterile semipermeable membrane (molecular weight cut-off: 3500 Da) was placed on the surface of a PDA plate. A 50 μL aliquot of DXQ-1 bacterial suspension was spread evenly over the membrane and incubated at 28 °C for 48 h. After incubation, the membrane was carefully removed, and a 5 mm diameter mycelial plug of *M. oryzae* was placed at the center of the plate. Pathogen growth was monitored, photographed, and recorded to assess inhibition.

### 2.5. Growth-Promoting Ability Test of DXQ-1

TP309 rice seeds were soaked in a 3% H_2_O_2_ solution for 12 h and then rinsed and soaked in sterile water until use. For bacterial culture, 10 mL of DXQ-1 fermentation broth was added to 1000 mL of full-nutrient solution for rice cultivation. Rice seedlings were grown in 96-well hydroponic boxes with rice full-nutrient solution as the control and maintained in a growth chamber under alternating light and dark conditions for 28 days. After cultivation, root length, leaf length, plant height, fresh weight, and dry weight were measured and analyzed.

### 2.6. Statistical Analysis

The means of three biological replicates were calculated, and differences between treatments were analyzed using Student’s *t*-test. Statistical significance was defined as *p* < 0.05.

## 3. Results

### 3.1. Identification and Characteristics of the Bacillus stercoris Strain DXQ-1

Through preliminary screening using the plate confrontation assay, five microbial strains exhibiting strong inhibitory effects against *M. oryzae* were identified (Appendix A). Among them, two strains—DXQ-1 and XQ2—demonstrated the most potent antagonistic activity and were selected for further evaluation using the drug-amended agar method. As shown in Appendix A, strain DXQ-1 exhibited the highest inhibitory effect, indicating its strong potential as a biocontrol agent against rice blast. As shown in Figure 1A, strain DXQ-1 formed milky white colonies with irregular edges on PDA medium, and its surface was covered by a wrinkled biofilm-like structure. Scanning electron microscopy revealed that the cells exhibited typical rod-shaped morphology. Furthermore, Gram staining analysis confirmed that DXQ-1 is a Gram-positive bacterium capable of forming spores. Based on phylogenetic analysis (Figure 1B), strain DXQ-1 was preliminarily identified as *Bacillus stercoris*.

To determine the optimal fermentation time for maximal antimicrobial activity of strain DXQ-1, its growth curve and the inhibitory effects of fermentation broths collected at different time points were assessed (Appendix A). DXQ-1 entered the logarithmic phase at 24 h and reached peak biomass at 60 h, followed by a stable stationary phase until 84 h. Antimicrobial activity was detectable as early as 24 h, despite low biomass (OD_600_ = 0.6). By 48 h, both biomass (OD_600_ = 2.0) and antimicrobial activity had increased significantly and approached a stable plateau. Considering the balance between growth and metabolite production, 48 h was identified as the optimal fermentation time to ensure high antimicrobial activity with efficient resource utilization. To optimize fermentation conditions for antimicrobial compound production by strain DXQ-1, basic media, pH, and temperature were systematically evaluated. Among the tested media, LB medium yielded the highest antimicrobial activity, with inhibition rates exceeding 90% against *M. oryzae* (Appendix A). Temperature tests showed that fermentation at 28 °C produced the most effective antimicrobial metabolites, with reduced activity observed above 33 °C or below 23 °C (Appendix A). pH screening revealed that DXQ-1 maintained strong inhibitory effects within a pH range of 5.0–9.0, with optimal activity at neutral pH (Appendix A). Thus, LB medium, neutral pH, and 28 °C were selected as optimal fermentation conditions.

### 3.2. Antagonistic Activity of the Biocontrol Bacterium B. stercoris Strain DXQ-1

To evaluate its antifungal activity, strain DXQ-1 was selected to investigate its effects on the mycelial morphology of *M. oryzae*. Scanning electron microscopy (SEM) observations of the mycelia following confrontation with DXQ-1 are presented in Figure 2. In the control group, *M. oryzae* exhibited typical cylindrical hyphae with smooth surfaces and intact, continuous structures (Figure 2A,C). In contrast, hyphae exposed to DXQ-1 displayed marked morphological alterations, including densely wrinkled surfaces, localized swelling at hyphal tips, and, in some cases, hyphal breakage and fragmentation (Figure 2B,D). These observations suggest that bioactive metabolites produced by DXQ-1 may disrupt cell wall synthesis and compromise structural integrity, thereby significantly inhibiting fungal growth. These findings further support the strong antifungal potential of strain DXQ-1 through structural interference with pathogenic hyphae.

The results of the detached leaf assay evaluating the biocontrol efficacy of DXQ-1 fermentation broth against rice blast are presented in Figure 3A,B. As the concentration of the fermentation broth increased, lesion length caused by *M. oryzae* infection progressively decreased. Notably, all treatment groups exhibited significantly reduced longitudinal lesion expansion compared to the untreated control group, suggesting that metabolites produced by strain DXQ-1 may effectively inhibit hyphal growth and spore dispersal by interfering with the pathogen’s infection process or modulating host defense responses. The antimicrobial substances in the fermentation broth of strain DXQ-1 significantly inhibited spore germination and appressorium formation of *M. oryzae* Guy11 in a concentration-dependent manner (Appendix A, Figure 3C). Undiluted broth almost completely suppressed germination and entirely blocked appressorium formation within 24 h. Even at 10-fold dilution, appressorium formation was minimal (6.25%). At 100-fold dilution, inhibition was reduced, with germination and appressorium formation reaching 100% and 98.75%, respectively, by 24 h. These results highlight the strong antifungal potential of DXQ-1 metabolites. Furthermore, when mixed suspensions of *M. oryzae* spores and various concentrations of DXQ-1 fermentation broth were sprayed onto rice plants under field conditions, disease symptoms observed one week later (Figure 3D,E) showed a marked reduction in lesion formation on treated leaves. These results indicate that strain DXQ-1 and its fermentation metabolites exhibit strong biocontrol potential against rice blast not only in vitro but also under natural field conditions.

### 3.3. Active Antimicrobial Substance Analysis of the Strain

The antimicrobial substances in DXQ-1 fermentation broth were extracted using solvents of varying polarity, and their bioactivity was assessed via the drug-amended plate method (Figure 4A). Among the extracts, the petroleum ether fraction exhibited the strongest inhibitory effect, significantly reducing pathogen colony diameter. The n-butanol extract showed moderate activity, while dichloromethane, aqueous, and ethyl acetate extracts displayed minimal inhibition, suggesting that the active compounds are non-polar and preferentially soluble in petroleum ether. To estimate the molecular weight of the active metabolites, a dialysis membrane assay was performed (Figure 4B). After three days of incubation, the *M. oryzae* colony growth was inhibited in the treatment group, indicating that the antimicrobial substances are capable of passing through the membrane. These results suggest that the active compounds have a molecular weight below 3500 Da.

Non-targeted metabolomic analysis of DXQ-1 fermentation broth was performed using LC-MS and the identified metabolites. The major compound classes included lipids and lipid-like molecules (35.83%), organic acids and derivatives (22.15%), benzene ring-containing organic compounds (11.40%), and organic heterocyclic compounds (11.07%). These findings suggest that DXQ-1 produces a chemically diverse array of metabolites, with strong representation of lipid-related and aromatic compounds. The top ten differential metabolites with the most significant abundance changes are listed in Table 1, including Leu-Pro, LPE 15:0, LPG 15:0, Propionyl-L-carnitine, and several aromatic or heterocyclic compounds, such as 4′-ethyl-N-(1-ethynylcyclohexyl)[1,1′-biphenyl]-4-carboxamide.

To assess the broad-spectrum antimicrobial potential of strain DXQ-1, its fermentation broth was tested against several phytopathogenic fungi using the drug-amended plate method. Target pathogens included *Magnaporthe oryzae*, *Rhizoctonia solani*, *Fusarium graminearum*, *Bipolaris maydis*, and *Nigrospora oryzae* (Figure 5). DXQ-1 exhibited the strongest inhibitory effects against *M. oryzae* and N. oryzae, nearly completely suppressing their growth. Notable inhibition was also observed against B. maydis and R. solani, while moderate suppression was recorded for *F. graminearum*. These results suggest that DXQ-1 possesses broad-spectrum antifungal activity, with particularly strong effects against rice and dove tree pathogens.

### 3.4. Evaluation of the Growth-Promoting Effect of Strain DXQ-1

The phosphate-solubilizing ability of strain DXQ-1 was evaluated for both inorganic and organic phosphorus sources. On PVK medium, DXQ-1 formed a distinct, clear halo around the colonies (Figure 6A), indicating effective inorganic phosphate solubilization. Furthermore, its robust growth on organic phosphorus-selective medium suggested a strong capacity to degrade organic phosphorus. Nitrogen fixation was assessed using Ashby nitrogen-free medium. As shown in Figure 6A, DXQ-1 exhibited vigorous colony growth, confirming its nitrogen-fixing ability. Siderophore production was evaluated using CAS agar, where DXQ-1 formed a clear orange-yellow halo, indicating its ability to secrete siderophores and chelate iron ions. Inoculated plants showed no signs of growth inhibition; instead, they exhibited enhanced morphological development (Figure 6B). Quantitative measurements (Figure 6C, Appendix A) revealed that plant height increased by 17.7% (44.37 cm vs. 37.69 cm), fresh biomass by 53.3% (0.23 g vs. 0.15 g), and dry biomass by 33.3% (0.04 g vs. 0.03 g) compared to the non-inoculated control group. All parameters showed significant improvements, demonstrating that DXQ-1 effectively promotes rice growth through multiple plant-growth-promoting mechanisms, including phosphate solubilization, nitrogen fixation, and siderophore production.

## 4. Discussion

Fungal diseases represent a major threat to global crop yields, and the overuse of chemical fungicides has raised serious concerns regarding environmental pollution, pathogen resistance, and food safety [27,28]. As a result, the development of efficient, eco-friendly biological control strategies has become increasingly vital for sustainable agriculture. *Bacillus* species are widely recognized for their broad-spectrum antimicrobial activity, primarily through the secretion of secondary metabolites that disrupt pathogen cell structures [1,5,21]. Additionally, they can induce systemic resistance in host plants by activating defense-related enzymes and releasing volatile compounds that inhibit fungal growth. As plant-growth-promoting rhizobacteria (PGPR), *Bacillus* strains also contribute to improved crop productivity and soil health [17,20,29]. In this study, we investigated the biocontrol potential of a self-isolated *Bacillus* strain, DXQ-1, with a particular focus on its inhibitory effects against key phytopathogens, such as *M. oryzae* and *R. solani*. By elucidating the antimicrobial mechanisms of DXQ-1, optimizing its fermentation conditions, and evaluating its plant-growth-promoting traits, this research aimed to characterize the functional properties and pathogen interaction mechanisms of DXQ-1 secondary metabolites and expand the available resources of biocontrol strains. Ultimately, this study lays the foundation for the development of a scalable production and application system for DXQ-1, providing a promising biopesticide for environmentally sustainable fungal disease management in crops.

In this study, a highly antagonistic strain, DXQ-1, was isolated using the dual culture plate method and identified as *B. stercoris* based on morphological characteristics, 16S rRNA sequencing, and phylogenetic analysis. DXQ-1 inhibited the growth of *M. oryzae* by suppressing spore germination and inducing severe hyphal deformations, including shrinkage and breakage. These findings are consistent with previous reports on the antifungal mechanisms of *Bacillus* species. For example, *B. velezensis* has been shown to produce broad-spectrum antifungal compounds, such as bacillomycin and surfactin, while other studies have linked cyclic lipopeptides like iturin A and fengycin to the disruption of fungal hyphae [27,28,30,31]. Similarly, DXQ-1 likely exerts its effects through such lipopeptides, as evidenced by the observed morphological damage and inhibition of appressorium formation in *M. oryzae*. Notably, LC-MS-based metabolomic analysis revealed that DXQ-1 produces abundant lipids and organic acids, further supporting the presence of active lipopeptides. Petroleum ether extracts of the fermentation broth showed the strongest antifungal activity, aligning with reports that lipophilic lipopeptides like surfactin and fengycin disrupt fungal membranes [32,33,34]. Additionally, DXQ-1 exhibits siderophore production and nitrogen fixation, suggesting a multifunctional role in both disease suppression and plant growth promotion. Collectively, these results underscore the promise of DXQ-1 as a broad-spectrum biocontrol agent with practical potential for sustainable crop protection.

Strain DXQ-1 demonstrated multiple plant-growth-promoting traits and biocontrol functions. It formed a distinct phosphate-solubilizing halo on PVK medium and exhibited notable organic phosphorus degradation capacity. This is consistent with findings that *B. amyloliquefaciens* dissolves inorganic phosphate through the secretion of organic acids [35,36], suggesting a similar solubilization mechanism in DXQ-1. Nitrogen-fixing ability was confirmed using Ashby medium, aligning with studies showing that *B. subtilis* enhances nitrogenase activity and nitrogen uptake [37,38]. Additionally, *B. subtilis* reduces NH_4_^+^-N retention and NH_3_ volatilization in alkaline soil by enhancing nitrogen sink capacity [39], further supporting DXQ-1′s nitrogen-fixing potential. Siderophore production was verified through the CAS assay, indicating DXQ-1′s ability to alleviate iron deficiency stress in plants. This trait not only facilitates plant growth but also contributes to pathogen suppression through iron competition [40]. Siderophore-mediated competition between *Staphylococcus aureus* and *Pseudomonas aeruginosa* significantly influences the severity of microbial infections [41], reinforcing the relevance of DXQ-1′s siderophore production to its antifungal efficacy. Furthermore, inoculation with DXQ-1 significantly improved rice plant height, fresh biomass, and dry biomass, consistent with previous reports [42,43]. In addition to its plant-growth-promoting functions, DXQ-1 also holds promise for food safety applications. *Bacillus* strains can degrade mycotoxins, such as deoxynivalenol and zearalenone [44,45], suggesting that DXQ-1’s metabolic capabilities may extend to mycotoxin mitigation. These findings support the potential of DXQ-1 as part of a synergistic strategy to combat resistance and minimize chemical inputs.

Overall, the results comprehensively elucidate the antagonistic mechanisms, metabolic characteristics, and biocontrol potential of DXQ-1. Its broad-spectrum antifungal activity, stability, and multifunctional traits make it a promising candidate for sustainable crop protection. Based on the existing literature, *Bacillus*-based biocontrol agents offer notable advantages in terms of gene cluster diversity, synergistic metabolite action, and formulation adaptability. Future research should focus on unraveling the molecular mechanisms of DXQ-1’s bioactive compounds and optimizing its industrial-scale production and field application.

## 5. Conclusions

In this study, strain DXQ-1 was identified as *Bacillus stercoris* through morphological and molecular analyses. DXQ-1 demonstrated broad-spectrum antifungal activity against multiple plant pathogens, especially *N. oryzae* and *M.oryzae*, and exhibited several plant-growth-promoting traits. Under optimized fermentation conditions (LB medium, pH 7.0, 28 °C, 48 h), the strain maintained over 90% inhibition against *M. oryzae*, indicating enhanced production of bioactive compounds. The active metabolites, primarily petroleum ether, were soluble and <3500 Da in molecular weight. Metabolomic profiling further identified compounds, such as Leu-Pro and LPE 15:0, as key contributors to the antifungal activity, highlighting DXQ-1’s potential as an effective biocontrol agent. DXQ-1 still requires further validation under field conditions to assess its antifungal effect and growth-promoting performance, as well as its potential impact on the agro-ecological environment. Long-term monitoring should be conducted to analyze its effects on soil microbial community structure and ecological balance after application in order to ensure its environmental friendliness. On this basis, it is recommended to develop more stable, cost-effective, and efficient formulations and to design composite microbial agents that combine disease prevention and nutrient enhancement functions, providing strong support for the sustainable development of green agriculture.

## Figures and Tables

**Figure 1 microorganisms-13-01538-f001:**
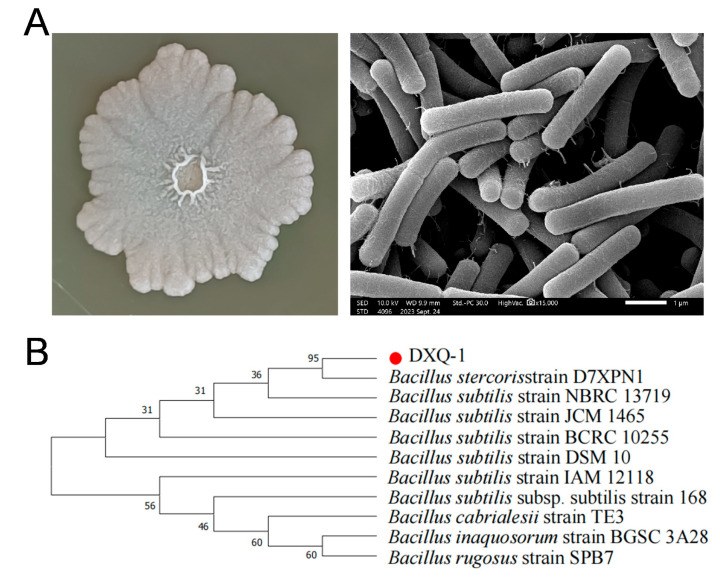
Identification and characteristics of *B. stercoris* strain DXQ-1. (**A**) The colony morphology, ultrastructure under the electron microscope, and bacterial staining of the of *B. stercoris* strain DXQ-1. (**B**) A phylogenetic tree was constructed based on the complete 16S rRNA gene sequences of strain DXQ-1 and other strains retrieved from GenBank. The tree was constructed using the neighbor-joining (NJ) algorithm, and 1000 bootstrap replicates were performed using MEGA 7.0 software. The bootstrap value (%) is indicated proximal to the nodes.

**Figure 2 microorganisms-13-01538-f002:**
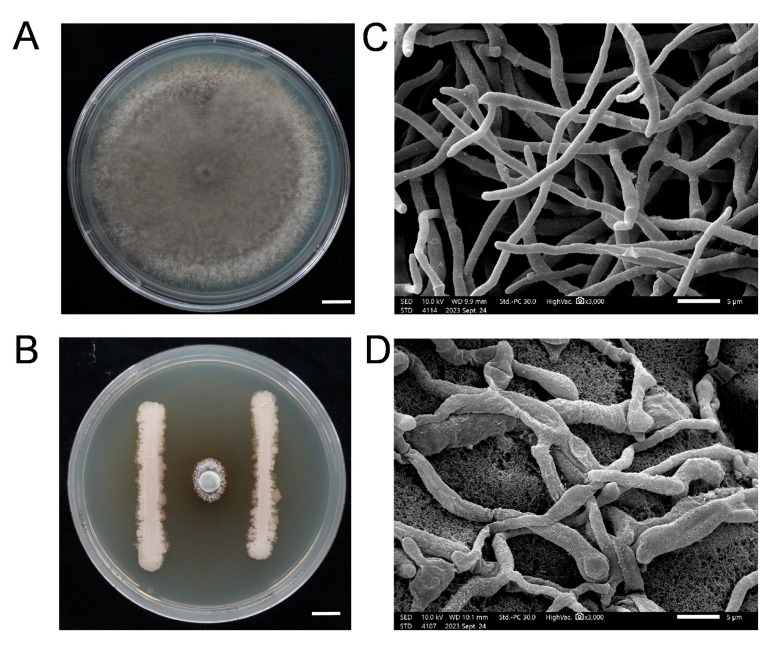
Antagonistic activity of the biocontrol bacterium *B. stercoris* strain DXQ-1 against the *M. oryzae* strain Guy11. (**A**) The mycelial morphology of *M. oryzae* strain Guy11. Scale bar = 1 cm. (**B**) DXQ-1 was inoculated into Guy11 colonies, and the colonies were photographed. (**C**) The ultrastructure of Guy11 in the control group; scale bar = 5 µm. (**D**) The ultrastructure of Guy11 after co-cultivation with *B. stercoris* strain DXQ-1; scale bar = 5 µm.

**Figure 3 microorganisms-13-01538-f003:**
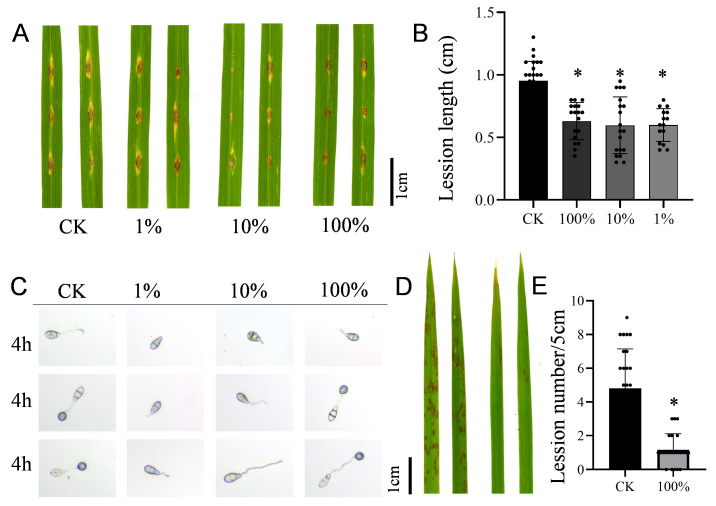
Effect of DXQ-1 extracts on disease incidence caused by *M. oryzae*. (**A**,**B**) Lesion symptoms and length after being incubated with Guy11, which included mixed fermentation broth and then *M. oryzae* spores 24 h later in the greenhouse. Rice leaves were incubated with 0 (CK), 1, 10, or 100% DXQ-1 fermentation broth. (**C**) Effect of DXQ-1 fermentation broth on spore germination of *M. oryzae*. Scale bar: 20 μm. (**D**,**E**) Lesion symptoms and blast disease lesion density after the therapeutic treatment, which involved spraying rice blast spores, followed by the fermentation broth 24 h later. Rice leaves were sprayed with DXQ-1 fermentation broth. Blast disease lesion density was quantified in infected leaf segments (5 cm in length) 5 days post-infection. The values are the means ± SDs. Asterisks denote a significant difference from the CK, determined using Student’s *t*-test (* indicates *p* < 0.05). n = 30 independent leaves in (**B**,**E**).

**Figure 4 microorganisms-13-01538-f004:**
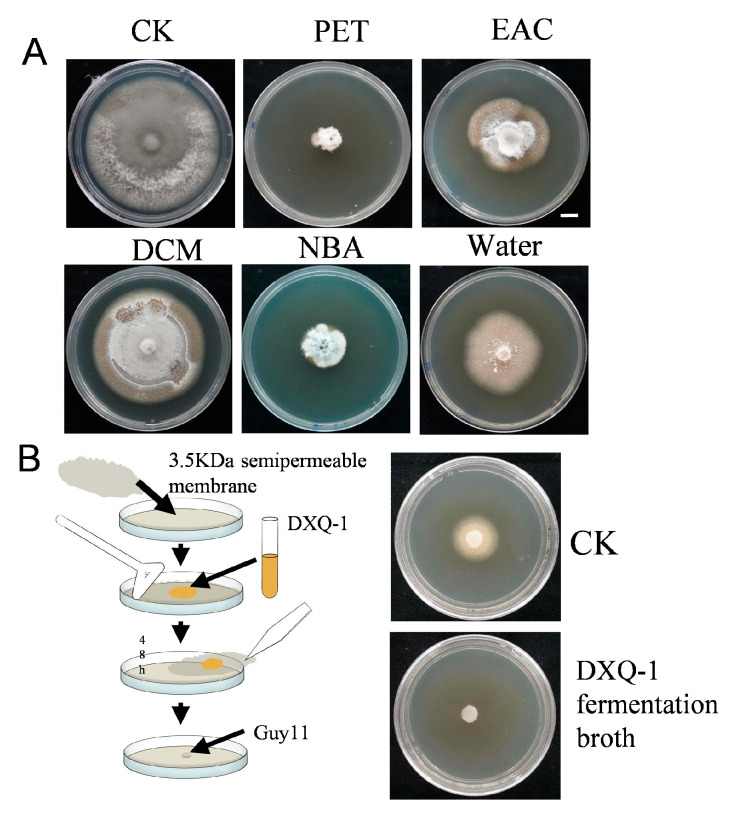
Active antimicrobial substance analysis of strain DXQ-1. (**A**) The mycelial morphology of Guy11 cultured on CM containing DXQ-1 extracts, petroleum extracts, dichloromethane extracts, n-butanol extracts, ethyl acetate extracts, and water extracts. PET: petroleum ether; EAC: acetic ester; DCM: dichloromethane; NBA: n-butanol. (**B**) Detection of the molecular weight of antibacterial substances using semipermeable membranes.

**Figure 5 microorganisms-13-01538-f005:**
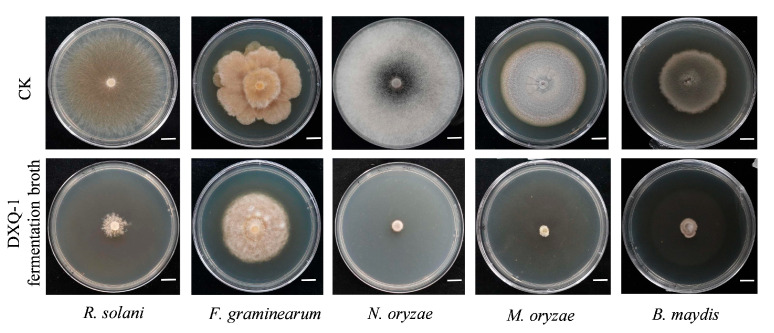
Antimicrobial spectrum of DXQ-1 extracts against phytopathogenic fungi. The mycelial morphology and colony diameters of *Rhizoctonia solani*, *Fusarium graminearum*, *Nigrospora oryzae*, *Magnaporthe oryzae*, and *Bipolaris maydis* were assessed on PDA plates supplemented with DXQ-1 extracts. The extracts were obtained from DXQ-1 cultured in PDB medium and incorporated into the plates. Plates without DXQ-1 extract served as controls (CK). Scale bar = 1 cm.

**Figure 6 microorganisms-13-01538-f006:**
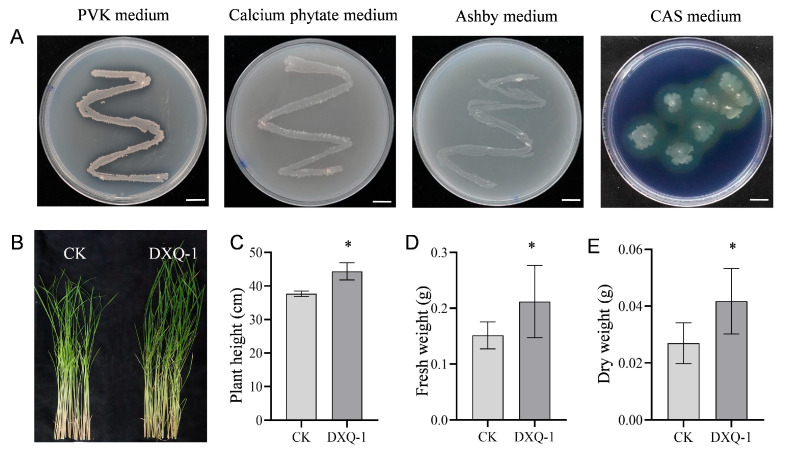
Plant-growth-promoting effects of DXQ-1. (**A**) Colony morphology of DXQ-1 cultured on PVK, calcium phytate, Ashby, and CAS media, indicating its abilities in phosphorus solubilization (PVK), organic phosphorus solubilization (calcium phytate), nitrogen fixation (Ashby), and siderophore production (CAS). (**B**) Phenotype, (**C**) plant height, (**D**) fresh weight, and (**E**) dry weight of TP309 rice plants after inoculation with DXQ-1 fermentation broth; sterile PDB served as the control. Data are presented as mean ± SD. Asterisks indicate statistically significant differences compared to the control (Student’s *t*-test, * *p* < 0.05). Scale bar in (**A**) = 1 cm.

**Table 1 microorganisms-13-01538-t001:** Analysis of DXQ-1 antifungal substances.

Compounds	*m*/*z*	Formula	DXQ-1 Fold Change
Leu-Pro	273.14	C_11_ H_20_ N_2_ O_3_	425.71
LPE 15:0	438.26	C_20_ H_42_ N O_7_ P	161.67
4′-ethyl-N-(1-ethynylcyclohexyl)[1,1′-biphenyl]-4-carboxamide	330.19	C_23_ H_25_ N O	135.95
LPG 15:0	469.25	C_21_ H_43_ O_9_ P	111.09
Propionyl-L-carnitine	216.12	C_10_ H_19_ N O_4_	98.36
1,1-Dimethyl-2-oxopropyl N-[2-(2-pyridyl)ethyl]carbamate	251.13	C_13_ H_18_ N_2_ O_3_	97.11
Esculin	339.07	C_15_ H_16_ O_9_	90.19
Lauric Acid	245.17	C_12_ H_24_ O_2_	89.65
Tanespimycin	566.28	C_31_ H_43_ N_3_ O_8_	83.86
2,3-Dinor prostaglandin E1	307.19	C_18_ H_30_ O_5_	64.45

## Data Availability

No new data were created or analyzed in this study. Data sharing is not applicable to this article.

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
