# Peer review of "Biocontrol Potential of Bacillus stercoris Strain DXQ-1 Against Rice Blast Fungus Guy11"

_microorganisms, 2025, doi:10.3390/microorganisms13071538_

Round 1
Reviewer 1 Report
Comments and Suggestions for Authors
The manuscript is interesting but needs re-organizing and careful checking. There are some mistakes (?) or at least inconsequences. Also many information which should be in Materials and Methods were also repeated in Results or exclusively in Results section.
In general, the Material and Methods subsections should match the subsections of Results.
Detailed remarks:
Lack of dots – f.ex. line37
Line 52 - what is Botrytis cinereal?
Lack of space between a word and bracket with references – whole manuscript
Line 116 -what kind of active substances F1–F4 do you think of?
Line 127 – where is section Section 2.2.1.1?
In the Abstract results from fields experiments were mentioned. However, there is no description of this field experiment in Materials and Methods
Lines 162 – 163 – but according to Table S1 there were two more strains with exactly the same inhibition rate as XQ2, so why you used XQ2?
Line 191 – latin species names should be in italics – check the whole manuscript
Line 199 – which “two” do you have in mind?
In line 216 you have mentioned detached leaf assay – while in Figure 3 A, B title capture there is field experiment mentioned- so what is true?
Line 219 – what “all treatments groups”? Where they are explained? They are mentioned in figure 3 caption – but it should be described in Material and methods also.
Figure 3 – where are the results of chemical fungicide isoprothiolane?
Figure 3DE – You have written: Rice leaves were sprayed with 0 (CK), 1, 10, or 100% DXQ-1 extract, so where are these results, there are four leaves (not described) and on 3E there is only CK and 100%.
Lines 278 and 281 – there is B. cinerea, while in the Figure 5 there is Bipolaris maydis?
Lines 300 – 302 – this should be in Material and methods described, not in Results
All this “Evaluation of the growth-promoting effect of strain DXQ-1” was not described in Material and methods
Line 387 – B. subtilis? – previously you have written that it was Bacillus stercoris - please be consistent in using latin names in the same manner through the whole manuscript.
Author Response
The manuscript is interesting but needs re-organizing and careful checking. There are some mistakes (?) or at least inconsequences. Also many information which should be in Materials and Methods were also repeated in Results or exclusively in Results section. In general, the Material and Methods subsections should match the subsections of Results.
Thank you for your thorough review and valuable feedback on our manuscript. We appreciate your detailed comments and suggestions. We will ensure consistency and coherence between the Materials and Methods subsections and the Results section, aligning them accordingly. Here is our point-by-point response to address the issues raised:
Detailed remarks:
Lack of dots – f.ex. line37
Response: sorry, we can’t find f.ex. in line37?
Line 52 - what is Botrytis cinereal?
Response: This was a writing error, and we have already made the correction. Thank you for your advise.
Lack of space between a word and bracket with references – whole manuscript
Response: We will address the lack of dots in the manuscript and correct spacing issues throughout the text.
Line 116 -what kind of active substances F1–F4 do you think of?
Response: This was an error in the statement, and we have already made the necessary corrections. We appreciate the reviewer's suggestions.
Line 127 – where is section Section 2.2.1.1?
Response: This was an error in the statement, and we have already made the necessary corrections. We appreciate the reviewer's suggestions.
In the Abstract results from fields experiments were mentioned. However, there is no description of this field experiment in Materials and Methods
Response: We have added the field experiment methodology to the Materials and Methods section as follows: For leaf inoculation in field condition, rice plants (cv. TP309) at the two-leaf stage were used. The fermentation broth was mixed with the spore suspension, and Tween-20 was added to a final concentration of 0.01%. The mixture was then sprayed onto rice seedlings that had been growing in the field for about 14 days. Disease incidence was recorded one week after inoculation. Each treatment was performed in triplicate.
Lines 162 – 163 – but according to Table S1 there were two more strains with exactly the same inhibition rate as XQ2, so why you used XQ2?
Response: Thank you for your thorough review. DXQ-1 and XQ-2 are both strains of Bacillus stercoris. While there are no significant differences in molecular biology and morphology between them, during the isolation process, XQ-2 exhibited slightly weaker antibacterial effects compared to DXQ-1. Therefore, we have chosen DXQ-1 as the focus for our further research.
Line 191 – latin species names should be in italics – check the whole manuscript
Response: Latin species names will be italicized consistently throughout the manuscript.
Line 199 – which “two” do you have in mind?
Response: We conducted transmission electron microscopy observations on M. oryzae and N. oryzae. However, as the results related to N. oryzae are not directly relevant to the focus of this study, we have removed that section. We appreciate the reviewer for pointing out our oversight in writing. Thank you for your understanding.
In line 216 you have mentioned detached leaf assay – while in Figure 3 A, B title capture there is field experiment mentioned- so what is true?
Response: We conducted in vitro inoculation assays and field spray tests. The legend for Figure 3 contained an error, which has now been corrected.
Figure 3. Effect of DXQ-1 extracts on disease incidence caused by M. oryzae. (A, B) Lesion symptoms and length after the incubated with Guy11, which mixed the fermentation broth then M. oryzae spores 24 h later in greenhouse. Rice leaves were incubated with 0 (CK), 1, 10, or 100% DXQ-1 fermentation broth. (C) Effect of DXQ-1 fermentation broth on spore germination of M. oryzae. Scale bar: 20 μm. (D, E) Lesion symptoms and blast disease lesion density after the therapeutic treatment, which involved spraying rice blast spores followed by the fermentation broth 24 h later. Rice leaves were sprayed with DXQ-1 fermentation broth. Blast disease lesion density was quan-tified in infected leaf segments (5 cm in length) 5 days post-infection. The values are the means ± SDs. Asterisks denote a significant difference from the CK as determined by Student’s t-test (* indicated p < 0.05). n = 30 independent leaves in (B, E).
Line 219 – what “all treatments groups”? Where they are explained? They are mentioned in figure 3 caption – but it should be described in Material and methods also.
Response: The treatments groups and their explanations have been detailed in both the figure captions and the Materials and Methods section.
Figure 3 – where are the results of chemical fungicide isoprothiolane?
Response: The results related to this section were not presented in the manuscript. We have removed the relevant statements. We apologize for any confusion and appreciate the reviewer's understanding.
Figure 3DE – You have written: Rice leaves were sprayed with 0 (CK), 1, 10, or 100% DXQ-1 extract, so where are these results, there are four leaves (not described) and on 3E there is only CK and 100%.
Response: In order to enhance the visual appeal of the figures, we have removed this section. The results are now presented as follows:
Lines 278 and 281 – there is B. cinerea, while in the Figure 5 there is Bipolaris maydis?
Response: This was a writing error that has been corrected. It should be Bipolaris maydis. Thank you for your careful review and valuable feedback.
Lines 300 – 302 – this should be in Material and methods described, not in Results
Response: we have removed this part and We have added the necessary information to the Materials and Methods section. Thank you for your valuable suggestions.
All this “Evaluation of the growth-promoting effect of strain DXQ-1” was not described in Material and methods
Response: We have added the necessary information to the Materials and Methods section. Thank you for your valuable suggestions.
Growth-promoting ability test of DXQ-1
TP309 rice seeds were soaked in a 3% H₂O₂ solution for 12 hours, then rinsed and soaked in sterile water until use. For bacterial culture, 10 mL of DXQ-1 Fermentation broth was added to 1000 mL of Full-nutrient solution for rice cultivation. Rice seedlings were grown in 96-well hydroponic boxes with rice full-nutrient solution as the control and maintained in a growth chamber under alternating light and dark conditions for 28 days. After cultivation, root length, leaf length, plant height, fresh weight, and dry weight were measured and analyzed.
Line 387 – B. subtilis? – previously you have written that it was Bacillus stercoris - please be consistent in using latin names in the same manner through the whole manuscript.
Response: This was a writing error that has been corrected. The consistency in using Latin names have been maintained throughout the manuscript.
Reviewer 2 Report
Comments and Suggestions for Authors
Dear Authors,
The manuscript "Biocontrol Potential of Bacillus stercoris strain DXQ-1 Against Rice Blast Fungus" presents very interesting aspect of using biological control against rice blast fungus. The control method is environmentaly friendly and needed by farmers. The manuscript correspondes with sustainable farming.
However, I have some comments which I have marked in the text. Especially you should describe Materials and Methods precisely. Some aspects are missed ad they should be completed. Results from field experiments should be presented in the table and should be describe in the Methods.
Discussion is well presented . conclusions should be improved. In the Section authors summarized the results but no conclusions included. References are choosen perfectly, a lot of the newest papers.
The paper has to be improved before publishing.

Author Response
The manuscript "Biocontrol Potential of Bacillus stercoris strain DXQ-1 Against Rice Blast Fungus" presents very interesting aspect of using biological control against rice blast fungus. The control method is environmentaly friendly and needed by farmers. The manuscript correspondes with sustainable farming.
However, I have some comments which I have marked in the text. Especially you should describe Materials and Methods precisely. Some aspects are missed ad they should be completed. Results from field experiments should be presented in the table and should be describe in the Methods.
Discussion is well presented . conclusions should be improved. In the Section authors summarized the results but no conclusions included. References are choosen perfectly, a lot of the newest papers.
The paper has to be improved before publishing.
Thank you for your detailed feedback on our manuscript "Biocontrol Potential of Bacillus stercoris strain DXQ-1 Against Rice Blast Fungus." We appreciate your positive comments on the use of biological control for rice blast fungus and its alignment with sustainable farming practices. We have carefully noted your suggestions and will address them as follows:
Please add the name of the fungus
Response: We have added the name of the fungus Guy11 in the title.
The results should be demonstrated in the table in the Results section.
Response: We have present the results in a table within the Results section.
Keywords should be different from the words present in the title.
Response: Keywords have been revised to be distinct from the words in the title. Keywords: Bacillus; Fungal pathogen; plant disease control; Magnaporthe oryzae ; anti-fungal metabolites
Line70-72 The authors should add aims of the work precisely.
Response: We have rephase the sentence as follows: This study identifies and characterizes a soil-derived B. stercoris strain, DXQ-1, with potent antifungal activity and plant growth-promoting effects, highlighting its potential as a sustainable biocontrol agent against rice fungal diseases. Our findings provide a scientific foundation for the development of robust, sustainable biocontrol strategies for integrated plant disease management.
The material and method section should be improve. The experiments are described very briefly, a lot of important information is missed.
Response: The Material and Methods section have been improved to include detailed descriptions of the experiments, including the medium used, colony isolation methods, plant growth effect test and identification of Bacillus strains.
Describe the method precisely. Which medium did you use, how did you isolate colonies ?
Response: we have described the method in material and method section as follows: Using the gradient dilution method, three concentration gradients of the soil suspension (10⁻¹, 10⁻², 10⁻³) were prepared with sterile water in a laminar flow hood. For each gradient, 200 μL of the soil suspension was evenly spread on the surface of Gauze’s synthetic broth medium, Luria-Bertani culture plate and Nutrient Agar (NA) medium, and Potato Dex-trose Agar (PDA) medium. Each treatment was done in ten plates.
How many isolates did you isolate and how many isolates did you examine. How did you identify Bacillus starins ?
Response: we have described the method in material and method section as follows:
Colonies with different morphologies were picked, streaked for purification, and num-bered. The purified single colonies were transferred into 50% glycerol and stored at -80°C for future use. We isolated over 300 colonies including Fungi, actinomycetes, Bacillus spp., etc. Using M. oryzae Guy11 as the indicator strain, biocontrol bacteria with antagonistic activity were preliminarily screened. After rescreening, the biocontrol Bacillus spp. with antagonistic effects were assigned corresponding identification numbers.
To further screen for Bacillus species, the soil suspension was heated at 60°C to induce spore formation, then spread onto NA medium. The plates were incubated upside down at 28°C for 3 days.
Line 91 Describe control treatment.
Response: we have described the method in material and method section as follows: The PDB mixed with PDA at 10% (v/v) was inoculated with M. oryzae strain Guy11 at the center, simultaneously with the control treatment.
Line 96 Please check the number. In GenBank under the number is Bacillus mojavensis strain MTC-8 .
Response: The correct GenBank number has been verified (PV789457). Thank you for bringing this to our attention.
Line 100 Add the source of the isolates. Add that the strains were also studied in the experiments.
Response: we have described the method in material and method section as follows: Plant pathogenic fungi including Magnaporthe oryzae (Guy11), Rhizoctonia solani (AG-1), Bipolaris maydis (C4), Fusarium graminearum (PH-1), and Nigrospora oryzae (LW11) were collected and are preserved in our laboratory.
Line114 Describe the experiment precisely. Was the EXperiment contuctes on all plants or only on the cut leaves ? how many leaves per one treatment ? Did you repeat the experiement ?
Response: we have described the method in material and method section as follows:
To observe conidial germination and appressorium formation, 30 μL of spore sus-pension was placed on a clean, hydrophobic slide and incubated in a humid chamber at 28 °C. 5–7 cm rice leaves (cv. LTH) at the two-leaf stage were wounded, treated with 6-BA (1 mg/L, pH 7.0), and inoculated with M. oryzae spore suspension (2×10⁵ conidia/mL) mixed with fermentation broth or sterile water. Inoculated leaves were incubated in dark-ness for 12 h, then under a 12 h light/dark cycle for 5–7 days. Lesion number and length were recorded and analyzed using ImageJ software[26]. Microscopic observations were performed using an Olympus optical microscope. For leaf inoculation in field condition, rice plants (cv. TP309) at the two-leaf stage were used. The fermentation broth was mixed with the spore suspension, and Tween-20 was added to a final concentration of 0.01%. The mixture was then sprayed onto rice seedlings that had been growing in the field for about 14 days. Disease incidence was recorded one week after inoculation. Each treatment was performed in triplicate.
Line 127Which section ?
Response: The writing error has been corrected. Thank you to the reviewer for pointing it out.
Line 129 How many plates for one treatment ?
Response: Six plates were incubated at 28 °C for 7 days, and colony diameters were measured. Each treatment was performed in triplicate.
Line 141 Which bacteriostatic substances were studied. The names of the substances are only in the sypplemtary Table 3, but they should be presented in the Section.
Response: Table of DXQ-1 antifungal substances were list in the result section. Esculin, Lauric Acid, Tanespimycin, 2,3-Dinor prostaglandin E1 were all test for its antifungal activity, while we did not get the positive results.
Add some information about statistical analysis of the results.
There are no information about experiments which results are presented in 3.4 Section - growth promoting effect of strain DXQ-1.
Response: we have described the method in material and method section as follows: Growth-promoting ability test of DXQ-1
TP309 rice seeds were soaked in a 3% H₂O₂ solution for 12 hours, then rinsed and soaked in sterile water until use. For bacterial culture, 10 mL of DXQ-1 Fermentation broth was added to 1000 mL of Full-nutrient solution for rice cultivation. Rice seedlings were grown in 96-well hydroponic boxes with rice full-nutrient solution as the control and maintained in a growth chamber under alternating light and dark conditions for 28 days. After cultivation, root length, leaf length, plant height, fresh weight, and dry weight were measured and analyzed.
Statistical analysis
The means of three biological replicates were calculated, and differences between treatments were analyzed using Student’s t-test. Statistical significance was defined as p < 0.05.
The data should be presented in the table. Statistical calculating ?
Response: We will ensure a precise description of the Materials and Methods section, including any missing aspects that need completion. Results from field experiments will be presented in a table and described in the Methods section.
The conclusions section should contained conclusions, future perpective of the methods and limitations of using the methods. Authors summarized the results.
The section should be improved.
Response: The conclusions section has been enhanced to include future perspectives and limitations of the methods as follows:
DXQ-1 still requires further validation under field conditions to assess its antifungal ef-fect and growth-promoting performance, as well as its potential impact on the agro-ecological environment. Long-term monitoring should be conducted to analyze its effects on soil microbial community structure and ecological balance after application, in order to ensure its environmental friendliness. On this basis, it is recommended to develop more stable, cost-effective, and efficient formulations, and to design composite microbial agents that combine disease prevention and nutrient enhancement functions, providing strong support for the sustainable development of green agriculture.
We appreciate your insightful comments and will work diligently to address these points to improve the manuscript before publication. Thank you for your valuable input.
Round 2
Reviewer 1 Report
Comments and Suggestions for Authors
Still some corrections from previous version not implemented:
line 55 - still Botrytis cinereal?